



# A spectral-structural characterization of European temperate, hemiboreal and boreal forests

Miina Rautiainen[1], Aarne Hovi[1], Daniel Schraik[1,2], Jan Hanuš[3], Petr Lukeš[3], Zuzana Lhotáková[4], Lucie Homolová[3]

[1]School of Engineering, Aalto University, Espoo, 00076, Finland
[2]Natural Resources Institute Finland, Helsinki, 00790, Finland
[3]CzechGlobe Global Change Research Institute of the Czech Academy of Sciences, Brno, 60300, Czech Republic
[4]Department of Experimental Plant Biology, Charles University, Prague, 12843, Czech Republic

*Correspondence to*: Miina Rautiainen (miina.a.rautiainen@aalto.fi)

**Abstract.** Radiative transfer models of vegetation play a crucial role in the development of remote sensing methods by providing a theoretical framework to explain how electromagnetic radiation interacts with vegetation in different spectral regions. A limiting factor in model development has been the lack of sufficiently detailed ground reference data on both structural and spectral characteristics of forests needed for testing and validating the models. In this data description paper, we present a dataset on the structural and spectral properties of 58 stands in temperate, hemiboreal and boreal European forests. It is specifically designed for the development and validation of radiative transfer models for forests but can also be utilized in other remote sensing studies. It comprises detailed data on forest structure based on forest inventory measurements, terrestrial and airborne laser scanning, and digital hemispherical photography. Furthermore, the data include spectral properties of the same forests at multiple scales: reflectance spectra of tree leaves and needles (based on laboratory measurements), forest floor (based on in situ measurements) and entire stands (based on airborne measurements), as well as transmittance spectra of tree leaves and needles and entire tree canopies (based on laboratory and in situ measurements, respectively). We anticipate that these data will have wide use in testing and validating radiative transfer models for forests and in the development of remote sensing methods for vegetation. The data can be accessed at:

Hovi et al. 2024a, https://doi.org/10.23729/9a8d90cd-73e2-438d-9230-94e10e61adc9 (for laboratory and field data) and

Hovi et al. 2024b, https://doi.org/10.23729/c6da63dd-f527-4ec9-8401-57c14f77d19f (for airborne data).

## 1 Introduction

Remote sensing of vegetation, and forests in particular, has experienced significant growth in recent years due to advancements in sensor technology, data processing and interpretation techniques, and new satellite missions (e.g., Fassnacht et al., 2024). At a global level, remote sensing can provide information about pressing global issues such as the connections between climate change and vegetation dynamics (e.g., Piao et al., 2020) and support for biodiversity conservation (e.g., Pettorelli et al., 2016).



Furthermore, at finer spatial scales, optical remote sensing allows detailed and accurate monitoring of, for example, vegetation
productivity, diversity and health (e.g., Kooistra et al., 2024; Hernández-Clemente et al., 2019).
Radiative transfer (RT) models of vegetation play a crucial role in the development of remote sensing methods by providing a
theoretical framework to explain how electromagnetic radiation interacts with vegetation in different spectral regions (Ross,
1981; Myneni & Ross, 1991). Based on mathematical formulations, these models allow us to understand and quantify the
complex interactions between radiation and canopy components, such as leaves, and stems, and the underlying soil (Liang,
2004). By modeling the radiative transfer processes, it is possible to explain the spectral signatures observed by remote sensing
instruments under different environmental and illumination conditions, or support future sensor design and planning of data
collection strategies (e.g., Vicent et al., 2015).
RT models and other physically-based canopy reflectance and transmittance models have been developed for over three
decades. For forests, these models (e.g., Gastellu-Etchegorry et al., 1996; North, 1996; Kuusk & Nilson, 2000; Leblanc &
Chen, 2001) are often more complicated, and require a larger number of input variables than models for other vegetation
ecosystems (e.g., Jacquemoud et al. 2009; Verhoef et al., 1984) due to the complex tree canopy architecture and subsequent
multiple interactions of photons both within and between canopy elements, and between forest floor and the canopy (e.g.,
Stenberg et al., 2008). Even though there are modeling approaches that require a smaller number of input variables for forests
(Stenberg et al., 2016), a limiting factor in model development has been the lack of extensive or sufficiently detailed ground
reference data on both structural and spectral characteristics of forests needed for testing and validating the models. This lack
of data affects both model developers and larger scientific frameworks, such as the RAdiation transfer Model Intercomparison
(RAMI) initiative (Gobron et al., 2023). While structural data on forests (e.g., tree height, crown length, number of trees per
ground area, canopy cover, leaf area index) are commonly available from sources such as forest inventory databases, spectral
data on forest components (e.g., leaf or forest floor reflectance and/or transmittance spectra) are less frequently accessible. In
addition, some structural properties (e.g., clumping index) that are relevant for RT models are also not commonly available
but can be derived from detailed structural measurements.
To date, major efforts in collecting ground reference data that can be used in radiative transfer models for forests have focused
on the North American continent. For instance, projects like the National Ecological Observatory Network (NEON) (NEON,
2024) and the Boreal Ecosystem-Atmosphere Study (BOREAS) (Sellers et al., 1997) offer input data for developing RT
modeling for forests. While these initiatives have primarily aimed to understand ecosystem dynamics, their datasets also
include key variables needed for RT models. For testing and validating forest RT models in European forests, there is only a
small number of datasets that include the necessary structural and spectral information across various scales (e.g., Kuusk et
al., 2009; Widlowski et al., 2015; Schneider et al., 2017; Liu et al., 2023). Furthermore, these datasets are limited in size,
containing information on only a few forest stands. Even though various solutions have been suggested to overcome the lack
of input data for RT models by using data from multiple sources (e.g., Malenovský et al., 2019), the lack of missing primary





data persists. In addition to having to collect the data from multiple sources representing different time periods or geographical
locations, these datasets are often not openly available according to FAIR Data principles (Wilkinson et al., 2016).
In this data description paper, we present a unique, open dataset on the structural and spectral properties of 58 stands in
temperate, hemiboreal and boreal European forests collected in a project funded by the European Research Council. The
dataset is specifically designed for the development and validation of radiative transfer models for forests but can also be
utilized in other remote sensing studies. It comprises detailed information on forest structure based on forest inventory
measurements, terrestrial and airborne laser scanning, and digital hemispherical photography. Furthermore, the dataset
includes spectral properties of the forests at multiple scales: reflectance spectra of tree leaves and needles (based on laboratory
measurements), forest floor (based on in situ measurements) and entire stands (based on airborne measurements), as well as
transmittance spectra of tree leaves and needles and entire tree canopies (based on laboratory and in situ measurements,
respectively). For distributing the data, we selected open, widely available formats.



## 2 Data collection

### 2.1 Study sites

We collected data from 58 forest stands representing different forest structures and species compositions in temperate, hemiboreal and boreal forests of Europe during summers 2019-2021 (Table 1, Fig. 1). The sites in Finland and the Czech Republic (Hyytiälä, Lanžhot, Bílý Kříž) are part of the Integrated Carbon Observation System (ICOS) which means that time series of meteorological and other ecosystem data are also openly available. The site in Estonia (Järvselja) also has a tower system for measuring variables related to atmosphere-biosphere interactions, and the data are available, per request from the tower manager. We have summarized information on the study sites in Table 1 and we provide a short verbal description of them in the following text.

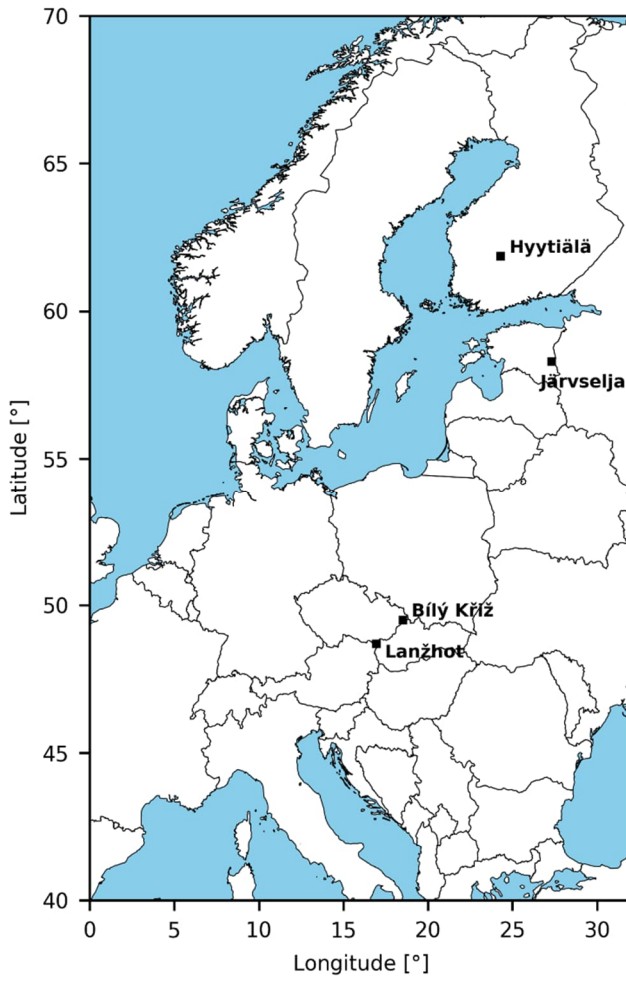

**Figure 1.** A map showing the locations of the study sites.



Our boreal study site was located in Finland, Hyytiälä (61°51'N, 24°18'E), and is a moderately flat (130–200 m a.s.l.) area
dominated by coniferous tree species. The forest floor is dominated by dwarf shrubs, graminoids, mosses or lichens. Bare soil
is rarely visible. Field measurements in Hyytiälä were conducted during 2019 and 2021.
Our hemiboreal site was located in Estonia, Järvselja (58°17'N, 27°19'E), and is a flat (30-45 m a.s.l.) area with mixed
broadleaved and coniferous forests. The forest floor is dominated by shrubs, dwarf shrubs, graminoids and mosses. Bare soil
is rarely visible. Field measurements in Järvselja were conducted during 2020.
Our temperate study sites, Lanžhot and Bílý Kříž, were located in the Czech Republic. Lanžhot (48°41'N, 16°57'E), is a
temperate broadleaf-dominated floodplain forest area (ca 150 m a.s.l.). The forest floor is sparsely covered by graminoids and
shrubs, and decomposed plant materials (or bare soil) is commonly visible due to a high game density. Bílý Kříž (49°30'N,
18°32'E), on the other hand, is a temperate coniferous mountain forest area (700–950 m a.s.l.) where the forest floor is
dominated by dwarf shrubs, graminoids and mosses. Field measurements in the Czech sites were conducted during 2019.
**Table 1.** Summary of the study plots and measurement campaigns.

|  | Hyytiälä | Järvselja | Bílý Kříž | Lanžhot |
|---|---|---|---|---|
| Forest biome | boreal | hemiboreal | temperate | temperate |
| Number of plots | 28 | 13 | 7 | 10 |
| Mean (and range) of tree height [m] | 20 (6 – 34) | 19 (4 – 39) | 23 (5 – 43) | 31 (18 – 40) |
| Mean basal area (and its range) [$m^2\ ha^{-1}$] | 23 (4 – 46) | 19 (4 – 51) | 34 (3 – 66) | 33 (14 – 60) |
| Effective plant area index [$m^2\ m^{-2}$] | 1.9 (0.1 – 3.9) | 2.5 (0.4 – 6.3) | 2.9 (0.4 – 4.7) | 3.7 (2.1 – 5.3) |
| Time of field campaign | 17 June – 26 July 2019, 8 July - 5 August 2021 | 24 June – 19 July 2020 | 16 – 29 September 2019 | 3 – 12 September 2019 |
| Time of airborne campaign (date, local time) | 13 July 2019, 08:57-10:21 | 15 July 2019, 12:57-14:07 | 4 September 2019, 11:01-11:07 | 4 September 2019, 12:14-12:22 |
| Solar zenith angle during airborne measurements | 51-60˚ | 37-38˚ | 47-48˚ | 42˚ |


## 2.2 Overview of measurement campaigns

We established 28 plots in Hyytiälä, 13 in Järvselja, 10 in Lanžhot, and 7 in Bílý Kříž (Fig. 2). Each plot was located within a homogeneous forest stand with a minimum distance of 30 m from the plot center to the stand border, to ensure that uncertainties in geolocation would not impact the interpretation of commonly used medium spatial resolution optical satellite data. The same sampling and measurement protocols were applied in collecting field data in all study sites.

In all plots, we carried out forest inventory (Sect. 2.3.1) and terrestrial laser scanning (Sect. 2.3.2), took hemispherical photographs of the tree canopy (Sect. 2.3.3) and conducted spectral measurements and estimation of vegetation fractional cover of the forest floor layer (Sect. 2.3.4). In addition, we measured the spectral transmittance of tree canopies in a subset of plots (Sect. 2.3.5) and measured the reflectance and transmittance spectra of the foliage of dominant tree species in all study sites (Sect. 2.3.6). An airborne measurement campaign in all study sites was conducted to obtain contemporaneous hyperspectral (Sect. 2.4.1) and laser scanning data (Sect. 2.4.2). The same aircraft and instrumentation were used for the acquisition of airborne data in all measurement campaigns. The datasets are provided by Hovi et al. 2024a and Hovi et al. 2024b.

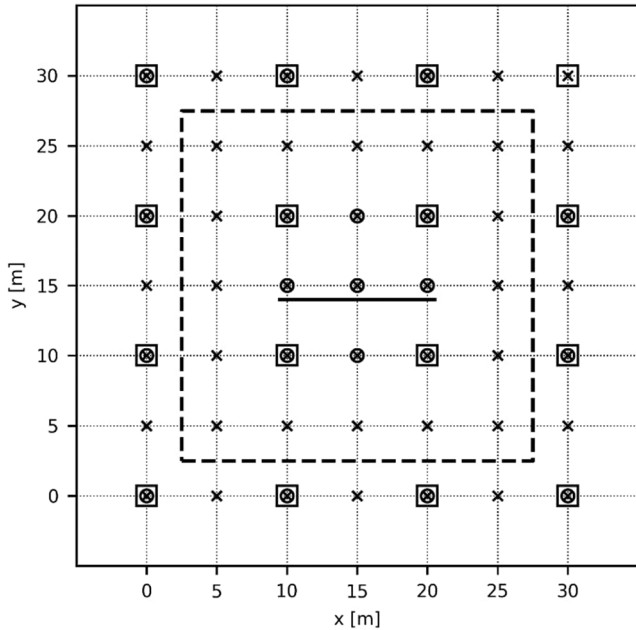

**Figure 2.** A diagram showing the sampling design for field measurements described in Section 2.3.



**2.3 Field datasets**

**2.3.1 Forest inventory**

We conducted forest inventory measurements to obtain detailed information on the tree species and stand structure and took photographs of each plot at six fixed locations to provide an overview of the forests for data users. Forest inventory was carried out with distinct protocols for mature stands (D > 10 cm) and young stands (D < 10 cm), categorized based on the average diameter at 1.3 m height (D) for trees. For simplicity, we refer to stands with D > 10 cm as mature stands and those with D < 10 cm as young stands.

In mature stands (n = 44), a tree-wise inventory was performed within a rectangular area measuring 25 m × 25 m (Fig. 2). The diameter at 1.3 m height was measured using a caliper, and the tree species were identified for every tree exceeding a predetermined diameter threshold. The thresholds were determined in relation to the average tree height in the plot (h) and were as follows: 8 cm if h < 16 m, 5 cm if 10 m ≤ h ≤ 16 m, and 2.5 cm if h < 10 m. Tree height was measured with a Vertex ultrasonic hypsometer for two trees (median trees of thickest 10% of trees) in each plot. These plots had 16 terrestrial laser scanning (TLS) points (see Section 2.3.2).

In young stands (n = 6), 16 circular sub-plots were measured, arranged in a 4 x 4 grid with a 10 m distance between grid points (see TLS grid in Fig. 2). The area of each sub-plot was 25 m² (i.e., had a radius of 2.82 m). Within each sub-plot, the number of trees per species, along with the diameter and height of a median tree per species, were measured. These plots had one terrestrial laser scanning (TLS) point (see Section 2.3.2).

An exception to the forest inventory protocol was made only in Hyytiälä for the plots (n = 8) measured in 2021, where relascope sampling was used to determine whether a tree belonged to the plot or not. Diameter at 1.3 m height was measured for all sampled trees, and tree height was measured for the median tree per species. These plots had one terrestrial laser scanning (TLS) point (see Section 2.3.2).

Descriptive forest characteristics were derived from the forest inventory data for each study plot. These include: number of stems per hectare, basal area, tree species proportions, and tree dimensions (i.e., stem diameter and tree height). More accurate description of the calculation of these variables is provided in the readme file of the data.

**2.3.2 Terrestrial laser scanning (TLS)**

We collected TLS data that can be used to characterize the 3D geometry of the forest canopies in all plots, comprising a total of ~2800 individual trees. The Leica P40 ScanStation, utilized in our study, operates at a wavelength of 1550 nm. It has a 6 mm beam diameter at the source and a 0.23 mrad beam divergence. The scan resolution equaled the beam divergence (i.e.,





0.23 mrad or around 0.013°). The only exceptions to this were the measurements 1) in Hyytiälä in 2021 (n = 8), and 2) in
young stands in Järvselja (n = 4) where the scan resolution was 0.31 mrad (0.018°). These exceptions are clearly labelled in
the dataset.
There were two alternative sampling strategies for collecting TLS data. The choice of sampling approach was based on stand
density at a height of 1-2 m above ground to avoid occlusion of co-registration targets, and time constraints. In 44 plots, TLS
scans were conducted at 16 grid points (Fig. 2), corresponding to the "mature" forest category described in Section 2.3.1. In
14 plots, TLS scans were conducted only at a single location (at the center of the plot, Fig. 2).
Scans were exclusively carried out under calm wind conditions (under 4 m s$^{-1}$ in 16-scan plots, under 8 m s$^{-1}$ in single scan
plots) and in dry weather. The scanning heights ranged from 1.4 to 1.8 m above the ground. In plots that had 16 scan positions,
co-registration of scans was done using 25 polystyrene sphere targets, mounted on 1.5-meter-tall sticks placed within the plot
area (Fig. 2). The co-registration errors were below 1 cm. All processing was done with the Leica Cyclone software.
The point clouds are intended for spatial modeling of canopy structure based on ray tracing rather than morphological
modeling. Therefore, no filtering was applied at any stage of the data processing to preserve information. The TLS data in
plots with 16 scans are available as full-resolution data, with each individual scan's point cloud stored separately, along with
the transformation parameters of the co-registration. For viewing purposes, we merged and downsampled the point clouds to
an average point spacing of 2 cm in Leica Cyclone, and cropped the plot to approximately a 35 m × 35 m area for the plots
that had 16 grid (scanning) points. For the single-scan plots, the downsampled point cloud includes all data. The downsampled
and merged point clouds are provided in the LAS format.

### 2.3.3 Hemispherical photographs

We also obtained a characterization of the tree canopies with hemispherical photography. Hemispherical photographs were
taken in each plot under diffuse illumination and windless or calm wind conditions with a Nikon D5000 digital camera
equipped with a geometrically calibrated lens (Sigma EX 4.5 mm f/2.8 DC HSM). The photographs were captured from 21
locations in each plot (Fig. 2) with the camera lens looking directly upwards. The camera was positioned at a height of 1.5 m
when the mean tree height in a stand was over 10 m, and at a height of 1.0 m in other forests.
The photographs were recorded in the best quality eight-bit JPEG format. We manually adjusted the exposure time based on
the illumination conditions and also took photographs with exposure times one stop higher and lower than the original, thus
doubling and halving the exposure time. In the processing of the photographs, we selected the one where the pixel values in
the blue band of the JPEG images filled the eight-bit dynamic range well without saturating the histogram, but also the other
photographs are included in the dataset.



These hemispherical photographs served as the basis for estimating effective plant area index (PAI$_{eff}$) and canopy gap fractions
in different view angles. Initially, the JPEG photographs were binarized according to Nobis and Hunziker (2005). Next,
effective PAI was calculated based on gap fractions determined for five concentric rings, each with median zenith angles of
10.7°, 23.7°, 38.1°, 52.8°, and 66.6°. This method closely followed the one presented in the manual of the LAI-2200 Plant
Canopy Analyzer (LI-COR 2012), with minor variations in the zenith angles (as listed above).

### 2.3.4 Hyperspectral measurements and other characteristics of the forest floor

We measured the spectral properties of the forest floor and estimated the fractional cover of different components forming the
forest floor in all plots. The composition of the forest floor ranged from nearly bare soil or litter to dense green vascular or
moss vegetation.
Hemispherical-conical reflectance factors (HCRF) of the forest floor (ranging from 350 to 2500 nm) were measured in a central
location in each plot using an Analytical Spectral Devices (ASD) FieldSpec4 spectrometer (serial number 18456) with a 25°
field-of-view. The initial spectral resolution ranged from 3 nm (for wavelengths ≤1000 nm) to 10 nm (for wavelengths >1000
nm), the sampling interval was 1.4 nm and 1.1 nm for visible and near infrared (VNIR), and shortwave infrared (SWIR),
respectively, and the instrument interpolated and outputted the data at 1 nm intervals. Please note that the same details on
spectral resolution also apply to the data measured by the spectrometers described later in Sections 2.3.5 and 2.3.6.
Measurements were consistently conducted under diffuse illumination conditions, so that the influence of unstable illumination
conditions on the forest floor (i.e., sun flecks, shadows) could be avoided and the data collected at different latitudes and times
of the day would be comparable. Preparations for the measurements included a warming-up period for the spectrometer lasting
at least 30 minutes.
In each plot, we established a 11-meter-long East-West oriented transect and made a total of 15 measurements at approximately
80 cm intervals along it (Fig. 2). Measurements were recorded in the nadir direction from a height of approximately 1.3 m. For
calibration, white reference measurements of a 25 cm × 25 cm Spectralon panel (with a nominal reflectance of 99%) were
conducted at both ends of the transect as well as at every third measurement point along the transect. Dark current
measurements were taken at both ends of the transect. The integration time, offset and gain of the spectrometer were adjusted
based on illumination conditions using automatic optimization.
Raw radiation signals (i.e., digital numbers, DN) were processed into hemispherical-conical reflectance factors (HCRF), and
the 15 pointwise measurements were averaged to produce a single spectrum per forest plot. We calculated the HCRF for each
measurement point by dividing the DN value of the forest floor by the DN value of the Spectralon panel and multiplied this
ratio with the reflectance of the white reference panel. Dark current readings were subtracted from all DN values prior to the
calculation. Because white reference readings were made at every third measurement point, we performed a linear interpolation



(in time) of the white reference measurements to obtain a value for all measurement points. The preprocessed data are provided
in the csv format.
Fractional cover was defined as the fraction of ground covered by living or dead plant material or lichens in 1 m$^2$ vegetation
quadrats. Fractional cover was estimated for all plots from nadir-view RGB (red, green, blue) photographs (four per plot) taken
by a Nikon D5000 camera at every fourth spectral measurement point (at a height of 1.5 m) along the transect where spectral
measurements were made. A wooden frame of 1 m × 1 m was placed at these measurement points, and the entire frame
(vegetation quadrat) was included in the photograph. After field work, the photographs were processed to obtain estimates of
fractional cover. The frame in each photograph was superimposed with a 10 × 10 grid, where each grid cell represented 1% of
the total image area. The forest floor present in each grid cell was visually classified into one of the following classes: 1)
vascular plants, 2) non-vascular plants (i.e., mosses), 3) lichen, 4) intact plant litter, or 5) decomposed plant litter. The criterion
for selecting one of the classes was that it was the most abundant class in the grid cell. Finally, the fractional cover of each
class in the photograph was determined by aggregating the grid cell specific results, and the average fractional cover of each
forest floor class within a forest plot was determined by calculating the mean of fractional cover values across the four
photographs.

## 2.3.5 Hyperspectral measurements of canopy transmittance

We conducted measurements of spectral transmittance of tree canopies (ranging from 350 to 2500 nm) in 8 plots in Hyytiälä,
6 plots in Järvselja, 4 plots in Lanžhot, and 4 plots in Bílý Kříž. Spectral transmittance of a canopy was defined as the ratio of
below-canopy spectral radiation flux to above-canopy spectral radiation flux.
For these measurements, we used two FieldSpec3 or -4 spectrometers and two identical cosine receptors (diffuser type, model
A124505) manufactured by ASD. In each forest plot, spectral transmittance was measured at 49 locations (Fig. 2). The ASD
FieldSpec4 spectrometer (serial number 18456) was consistently employed for measurements within the forest (i.e., below-
canopy), whereas the ASD FieldSpec3 or -4 (serial number 18641 or 16089) served as reference spectrometer (i.e., above-
canopy). For the above-canopy measurements, a tripod was used to affix the cosine receptor which was measuring at 15 second
intervals in an open area within the study site (within <2 km distance from the plots). Measurements were conducted only
under cloud-free conditions, with solar elevation angles ranging from 30° to 45°.
Preparations for the measurements included a warming-up period for the spectrometers lasting at least 30 minutes, automatic
optimization of the spectrometers' integration time and gain settings, and an intercalibration of the two spectrometers. The
intercalibration took place at the beginning and end of each measurement period (max 3 h 20 min). It involved placing the
cosine receptors next to each other in an open area and conducting ten measurements, with each measurement comprising 30
averaged spectra from both spectrometers.



After the field campaign, the data were processed into canopy spectral transmittance ($T$) as
$$T = \frac{f_{bc}s_{bc}}{f_{ac}s_{ac}}k,$$ (1)
where $s_{bc}$ and $s_{ac}$ are raw signal (DN) values recorded below and above canopy, respectively, $k$ is the ratio of DNs measured
by the two spectrometers under identical irradiance conditions (obtained from the intercalibration measurements), and $f_{bc}$ and
$f_{ac}$ are correction factors that take into account possible changes of the integration time (at wavelengths up to 1000 nm) or the
detector gain (at wavelengths above 1000 nm) due to re-optimization of either of the spectrometers during the measurement
period. Re-optimization was needed if signal saturation occurred, for example, when measuring before noon, as the solar
irradiance increased towards noon. All quantities in the equation are wavelength- or detector-dependent.
**2.3.6 Hyperspectral measurements of tree leaves and needles**
We measured the directional-hemispherical reflectance factors (DHRF) and directional-hemispherical transmittance factors
(DHTF) ranging from 350 to 2500 nm of leaves and needles for fifteen dominant tree species within the study sites, adding up
to a total of 1314 samples. The two coniferous tree species that we sampled were Norway spruce (*Picea abies* (L.) H. Karst.)
and Scots pine (*Pinus sylvestris* L.). The thirteen broadleaved tree species that we measured were common hazel (*Corylus*
*avellana* L.) English oak (*Quercus robur* L.), European alder (*Alnus glutinosa* (L.) Gaertn.), European ash (*Fraxinus excelsior*
L.), European aspen (*Populus tremula* L.), European hornbeam (*Carpinus betulus* L.), European Turkey oak (*Quercus cerris*
L.), goat willow (*Salix caprea* L.), hedge maple (*Acer campestre* L.), littleleaf linden (*Tilia cordata* Mill.), silver birch (*Betula*
*pendula* Roth), white poplar (*Populus alba* L.) and willows (*Salix* sp.). For simplicity, we will refer to leaves and needles
collectively as foliage in the following text.
The foliage samples were measured in laboratory conditions using ASD RTS-3ZC integrating spheres which were equipped
with a 10 W collimated halogen light source. The integrating sphere was coupled with an ASD spectrometer (FieldSpec3 serial
number 16089, or FieldSpec4 serial number 18456 or 18641). Preparations for the measurements included a warming-up
period for the spectrometer lasting at least 30 minutes.
In all study sites, visibly healthy foliage samples were obtained from both sun-exposed positions in the top-of-canopy and
shaded positions in the bottom-of-canopy using professional tree climbers, towers or long pruning shears. After cutting a
branch from the tree, it was stored in a cool environment (with a maximum storage time of 12 hours), maintained with adequate
watering, and foliage was removed from the branch immediately before the spectral measurements.
For coniferous trees, two age cohorts of needles were always sampled: current year (c0) and one-year-old (c1) needles. In
Hyytiälä, Järvselja and Bílý Kříž, three trees representing each tree species were sampled, with three samples collected for
each foliage class in each tree. This means that for all tree species, we sampled sun-exposed c0 and shaded c0 foliage samples,



and for conifers, we also sampled sun-exposed c1 and shaded c1 foliage classes. For less common broadleaved species in
Järvselja (European ash, goat willow, littleleaf linden, common hazelnut, and unspecified willow), samples from one tree were
obtained, and three sun-exposed c0 leaves were collected per tree species. In Lanžhot, one to four trees were selected for
sampling. Each tree contributed one sample for every foliage class, including shaded c0 or sun-exposed c0.
For the duration of the spectral measurement of a sample in Hyytiälä, Järvselja and Bílý Kříž, the sample (i.e., a leaf or a set
of 7-10 needles) was fixed in a custom-made sample holder (see Fig. 1 in Hovi et al., 2020 for sample holder design) that was
then fastened to the integrating sphere. Needles were arranged in the sample holder with a spacing of 0.5–1 times the width of
a single needle (as recommended by Yáñez-Rausell et al. 2014), and leaves were placed so that major veins were not included
in the measured spot. In Lanžhot, leaves of broadleaved species were not attached to sample holders.
We conducted measurements of DHRF and DHTF on both sides of the sample (corresponding to adaxial and abaxial in
broadleaved species), along with white reference measurements for both reflectance and transmittance. A photon trap was used
in the reflectance measurements to assess stray light. Our white reference was a Spectralon panel with 99% nominal
reflectance. The raw data were processed to derive leaf or needle DHRF and DHTF for all samples. For brevity, we denote
DHRF with $R$ and DHTF with $T$ in the following equations:
$$R = \frac{s_R}{s_{ref,R}} \frac{1}{1-P_{gap,R}} R_{ref} , \qquad (2)$$
$$T = \left(\frac{s_T}{s_{ref,T}} - P_{gap,T}\right) \frac{1}{1-P_{gap,T}} R_{ref}, \qquad (3)$$
where $s_R$ and $s_T$ represent the raw signals (DN) obtained from the reflectance and transmittance measurements. Similarly, $s_{ref,R}$
and $s_{ref,T}$ denote the DNs from the white reference measurements for reflectance and transmittance, respectively. $R_{ref}$ indicates
the reflectance of the white reference panel, while $P_{gap,R}$ and $P_{gap,T}$ denote the gap fractions in the sample. Before $R$ was
computed, stray light was first subtracted from $s_R$ and $s_{ref,R}$.
For broadleaved species, the gap fraction was assigned a value of 0 in the above calculations. Coniferous samples, on the other
hand, included gaps between needles, and thus, we determined the gap fractions using a digital film scanner (Epson Perfection
V550, 800 dpi resolution). The detailed procedure for determination of gap fraction was done according to Hovi et al. (2020).
Finally, to address a slight inherent bias in transmittance measurements with the ASD RTS-3ZC integrating sphere (reported
by Hovi et al. 2020) and to ensure that the sum of DHRF and DHTF did not exceed one in the near-infrared (NIR) region, we
implemented an empirical correction in which the DHTF spectra were multiplied with a correction factor of 0.945.
For data users, we provide the spectra for all samples as well as analysis-ready datasets. The analysis-ready datasets contain i)
the mean DHRF and DHTF spectra and their standard deviations for all tree species, canopy positions (top and bottom), needle
age classes (c0, c1) and study sites, and ii) plot-specific mean DHRF and DHTF spectra which have been weighted based on
tree species and needle age class proportions (i.e., computed from i).





**2.4 Airborne datasets**

**2.4.1 Hyperspectral data**

We arranged flight campaigns in mid-July 2019 in Hyytiälä and Järvselja, and in early September 2019 in Lanžhot and Bílý Kříž (Table 1), representing green phenological conditions. Airborne hyperspectral measurements were collected across all study sites using the CASI-1500 and SASI-600 hyperspectral pushbroom sensors from Itres Ltd., Canada, mounted on a Cessna C208B aircraft which is part of the Flying Laboratory of Imaging Systems (FLIS) operated by the CzechGlobe Global Change Research Institute (Hanuš et al., 2023). The CASI-1500 covered visible (VIS) to NIR wavelengths (382 to 1052 nm), while the SASI-600 sampled NIR and shortwave-infrared (SWIR) wavelengths (958 to 2443 nm). Both sensors had a sampling interval and spectral resolution of 15 nm and underwent spectral and radiometric calibration prior to the flight campaigns in March 2019.

During the flight campaigns, the aircraft flew at an altitude of approximately 1 km above ground level. This yielded ground pixel sizes of 0.5 m (CASI) and 1.25 m (SASI). The CASI and SASI data were acquired in near-nadir observation geometry with a +/- 20° field-of-view. The flying azimuth direction closely matched the solar azimuth – the purpose of this was to reduce potential spectral differences within the same study site caused by reflectance anisotropy of forests in the solar principal plane. During acquisitions, the Sun zenith angle ranged from 37° to 60°, and flight lines overlapped by 60–80%.

The raw DN data from the hyperspectral sensors underwent initial radiometric correction with the RadCor software (version 11) produced by Itres Ltd. Subsequently, geo-orthorectification was performed using GeoCor (version 5.6). The data were ortorectified to a surface model, which represents the top-of-canopy in vegetated areas, and the ground elevation elsewhere. Atmospheric correction was carried out with the ATCOR-4 software bundle (version 7.2.0 or 7.3.0), employing a database of atmospheric look-up tables generated with the MODTRAN5 radiative transfer code. In this correction, sensor measurements were adjusted for path and adjacency radiances. Inflight radiometric (vicarious) calibration was conducted for each site using a known bright reflectance target. Spectral bands highly affected by water vapor in the atmosphere (i.e., 895-1003 nm, 1092-1168 nm, 1302-1528 nm, and 1737-2038 nm) were nonlinearly interpolated and depended on local atmospheric conditions. No topographic correction was applied. The data produced through this processing chain are provided as at-surface (also called top-of-canopy) hemispherical-directional reflectance factors (HDRF).

Finally, we inspected the CASI and SASI data manually to remove clouds or cloud shadows from areas corresponding to our study plots. During the flights, clouds were intermittently present over Hyytiälä site and occasionally in Bílý Kříž site. The Lanžhot and Järvselja flights, on the other hand, had cloudless conditions. Nearest-to-nadir cloud-free data from a 100 m × 100 m area around each plot were extracted and serve as an analysis-ready dataset. In addition, data from the entire study sites are provided. These data cover approximately 4 km × 4 km areas in Hyytiälä and Järvselja, 2 km × 3 km in Lanžhot, and 2 km × 2 km in Bílý Kříž.





**2.4.2 Laser scanning data (ALS)**

Airborne laser scanning (ALS) data were collected simultaneously with the airborne hyperspectral data using a Riegl LMS-Q780 laser scanner (Riegl Gmbh, Austria) mounted on the same Cessna aircraft. The laser scanner operated at a wavelength of 1064 nm, had a 0.25 mrad beam divergence, and a maximum scan zenith angle of 30°. The pulse density at the study plots was 48, 32, 10, and 9 pulses m$^{-2}$ in Hyytiälä, Järvselja, Lanžhot, and Bílý Kříž, respectively. The differences between sites stem from different overlap of flight lines. In Hyytiälä, the elevated pulse density was also partly due to repeated flight lines due to occasional cloud cover. The raw waveform data were processed into point cloud format using RiProcess (version 1.8.4), RiAnalyze (version 6.2.2), RiWorld (version 5.1.3), and GeoSysManager (version 2.0.8) software. We also computed raster digital elevation models with a pixel size of 1 m, by interpolating from the ground points classified with LASTools software. Similarly to the airborne hyperspectral data, analysis-ready data were extracted for a 100 m × 100 m area around each study plot, and the data are also provided for the entire study sites as original point clouds and denoised data. Denoised data were processed to filter out points originating from the sky (due to e.g., clouds) or false points under ground.

**2.5 External field datasets**

Field datasets from other sources, and relevant to physically-based remote sensing but not included in our campaigns, are available for the study sites. We have summarized these datasets in Table 2. They include 1) reflectance spectra of tree bark for boreal and temperate tree species, 2) additional data sets on optical properties of Norway spruce needles from the Czech study sites, and 3) forest meteorology, greenhouse gases, air quality and soil measurements from ICOS towers.

**Table 2.** Ancillary data sets relevant for RT modeling of forests available for the study sites from other projects.

| Description of data set | Source |
|---|---|
| Stem bark reflectance spectra for boreal and temperate tree species | DOI: 10.17632/pwfxgzz5fj.2 |
| Forest meteorology, greenhouse gases, air quality and soil measurements | |
| for Hyytiälä site | DOI: 10.23729/23dd00b2-b9d7-467a-9cee-b4a122486039 |
| for Lanžhot site | https://meta.icos-cp.eu/objects/LaXYKv7nUEOYLD62wr43PK7H (last access 11 April 2024) |
| for Bílý Kříž site | https://meta.icos-cp.eu/objects/Ru01KATyDlvqFkOzvB7eBcrY (last access 11 April 2024) |
| Optical properties of Norway spruce needles | DOI: 10.17632/vycrxc4vpz.1 |



## 3 Results

The data allow examining comprehensively the spectral and structural properties of forest stands. We summarized the different data sources in two sets of figures, using a coniferous stand from Bílý Kříž (Fig. 3) and a broadleaved stand from Hyytiälä (Fig. 4) as examples. These two forest stands illustrate the variation in structural and spectral properties both within and between stands present in the new dataset. For example, the point clouds produced by laser scanning sensors and described in this paper (Fig. 3c-d, 4c-d) can be used to visualize and compute canopy height distribution or density metrics, or to assess the spatial distribution patterns of trees or foliage clumping in the study stands. The variation in the spectral properties of the study stands, on the other hand, can be divided into several parts to examine tree leaf-level (Fig. 3e, Fig. 4e), forest floor level (Fig. 3f, Fig. 4f) and tree canopy level (Fig. 3g-h, Fig. 4g-h) phenomena. As a specific example of a key structural variable needed in RT modeling of vegetation, we publish data on tree canopy gap fractions in different view angles based on hemispherical photography. On average, in our coniferous stands, canopy gap fractions were approximately two times as high as in the broadleaved stands, and in both types of forests, the gap fractions decreased linearly towards the horizon (Fig. 5).

Using the datasets described in this paper, differences in the spectral properties of forests can be investigated at multiple scales (Fig. 6). In presenting the data here, we refer to the spectral regions as visible (~400–700 nm), near infrared (~700–1300 nm) and shortwave infrared (~1300–2500 nm). In both coniferous and broadleaved stands, the reflectances were notably higher at tree leaf level than at stand (canopy) level throughout the entire measured spectrum (Fig. 6a-b). Forest floor reflectances, on the other hand, were usually lower than tree leaf level reflectances but higher than canopy level reflectances in the visible and near-infrared regions. However, in the shortwave infrared region, the forest floor had, on average, a higher reflectance than tree leaves or canopies in coniferous stands, and a reflectance similar to that of tree leaves in broadleaved stands (Fig. 6a-b). An especially unique feature of this dataset is that also transmittance spectra at leaf and canopy levels were measured so that they could be used in, for example, testing the performance of RT models. In our data, the canopy level spectral transmittance of coniferous stands was more stable throughout the spectrum than the canopy level transmittance of broadleaved stands, and that transmittances at leaf and canopy levels were usually lower in our coniferous study plots than in broadleaved study plots (Fig. 6c-d). Furthermore, the data show that in the visible region, the spectral transmittance at canopy level was higher than the spectral transmittance at leaf level. In the near-infrared and shortwave infrared regions, on the other hand, leaf level transmittances were higher than canopy level transmittances. An exception to this was in the coniferous stands in two spectral regions – around 1400–1500 nm and above ~1900 nm – where canopy level transmittances were again higher than leaf level transmittances. In broadleaved stands, the canopy spectral transmittances in shortwave infrared were higher than leaf level transmittances only in a small region around 1900–2000 nm.

Finally, the data also allow examining relationships between structural and spectral properties of forests through a combination of contemporaneous airborne laser scanning and hyperspectral data (Fig. 7). These data can be used to illustrate, for example, that, in the visible spectral region, forest reflectance decreased as a function of increasing canopy cover (defined as the first



380    echo cover index in ALS data) across forest stands representing different biomes (Figs. 7a, c), but that in the near-infrared and

381    shortwave infrared regions, broadleaved and coniferous stands with closed canopies (i.e., high canopy cover values) formed

382    two distinct groups so that coniferous stands had notably lower HDRFs than broadleaved stands did (Figs. 7e, g). Similar

383    phenomena were also observed in the relationships between forest reflectance and canopy height (defined as the 95th percentile

384    of all canopy echoes) obtained from ALS data (Fig. 7b, d, f, h).





385

386      (figure caption on following page)



**Fig. 3.** A collection of figures summarizing the different types of data collected for a pure coniferous plot located in Bílý Kříž (stand ID "BK_SPRUCE2" in the dataset). The dominant tree species is Norway spruce (99% of basal area), effective plant area index 2.8, and mean tree height 20.8 m. **A.** An overview photograph of the plot (from the north-east corner towards the plot center). **B.** A hemispherical photograph of the canopy (Section 2.3.3). **C.** Point cloud visualization of the plot based on terrestrial laser scanning data from the south-west corner towards the plot center based on a downsampled point cloud (Section 2.3.2). **D.** Point cloud visualization of the plot based on airborne laser scanning data from the south-west corner towards the plot center (from view zenith angle 45°, 17 pulses m$^{-2}$) (Section 2.4.2). **E.** Mean leaf-level reflectance and transmittance spectra (DHRF and DHTF, respectively) and their standard deviations for current year and one-year-old needles of the dominant tree species in the plot (Section 2.3.6). **F.** Mean reflectance spectrum (HCRF) and its standard deviation for the forest floor in the plot (Section 2.3.4). Spectral regions with noise were caused by atmospheric water vapor. **G.** Mean spectral transmittance and its standard deviation for the tree canopy layer (Section 2.3.5). Spectral regions with noise were mainly caused by atmospheric water vapor, but also by the reduced sensitivity of the cosine receptor at the end of the spectral range (>2200 nm). **H.** Mean reflectance spectrum (HDRF) and its standard deviation for the entire plot (25 m × 25 m area) based on airborne measurements (Section 2.4.1).






(figure caption on following page)




**Fig. 4.** A collection of figures summarizing the different types of data collected for a broadleaved plot located in Hyytiälä
(stand ID "HY_BIRCH2" in the dataset). The dominant tree species is silver birch (85% of basal area), effective plant area
index 1.5, and mean tree height 23.2 m. **A.** An overview photograph of the plot (from the north-west corner towards the plot
center). **B.** A hemispherical photograph of the canopy (Section 2.3.3). **C.** Point cloud visualization of the plot based on
terrestrial laser scanning data from the south-west corner towards the plot center based on a downsampled point cloud (Section
2.3.2). **D.** Point cloud visualization of the plot based on airborne laser scanning data from the south-west corner towards the
plot center (from view zenith angle 45°, 48 pulses m$^{-2}$) (Section 2.4.2). **E.** Mean leaf-level reflectance and transmittance spectra
(DHRF and DHTF, respectively) and their standard deviations for the dominant tree species in the plot (Section 2.3.6). **F.**
Mean reflectance spectrum (HCRF) and its standard deviation for the forest floor in the plot (Section 2.3.4). Spectral regions
with noise were caused by atmospheric water vapor. **G.** Mean spectral transmittance and its standard deviation for the tree
canopy layer (Section 2.3.5). Spectral regions with noise were mainly caused by atmospheric water vapor, but also by the
reduced sensitivity of the cosine receptor at the end of the spectral range (>2200 nm). **H.** Mean reflectance spectrum (HDRF)
and its standard deviation for the entire plot (25 m × 25 m area) based on airborne measurements (Section 2.4.1).

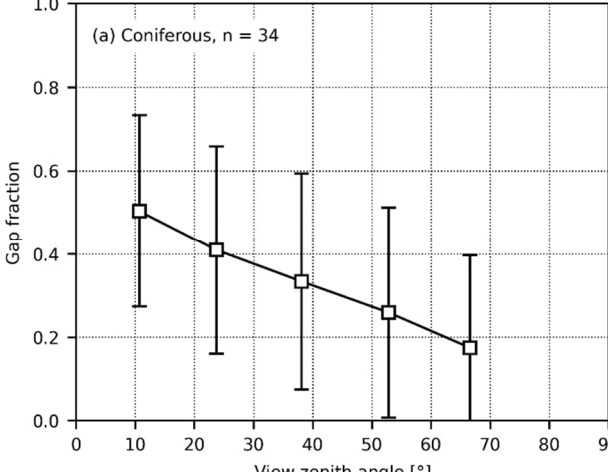
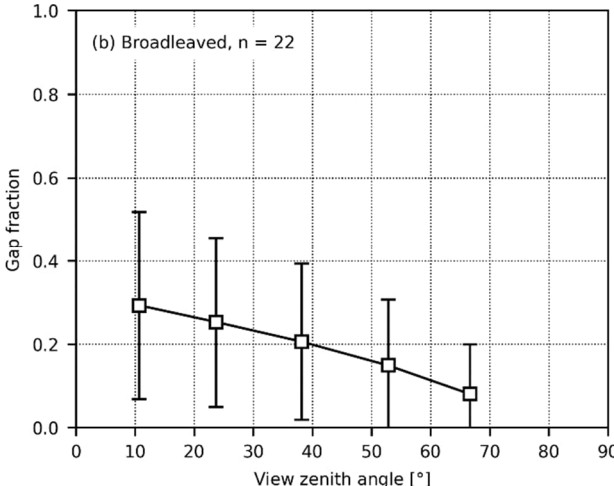


**Fig. 5.** Mean and standard deviation of canopy gap fractions in concentric view zenith angles as obtained from hemispherical
photographs in **A)** coniferous and **B)** broadleaved forests. Here, coniferous and broadleaved forests were defined so that at
least 75% of the trees (based on basal area) within the plot were coniferous or broadleaved species, respectively. The data
shown in this figure are based on measurements described in Section 2.2.3.

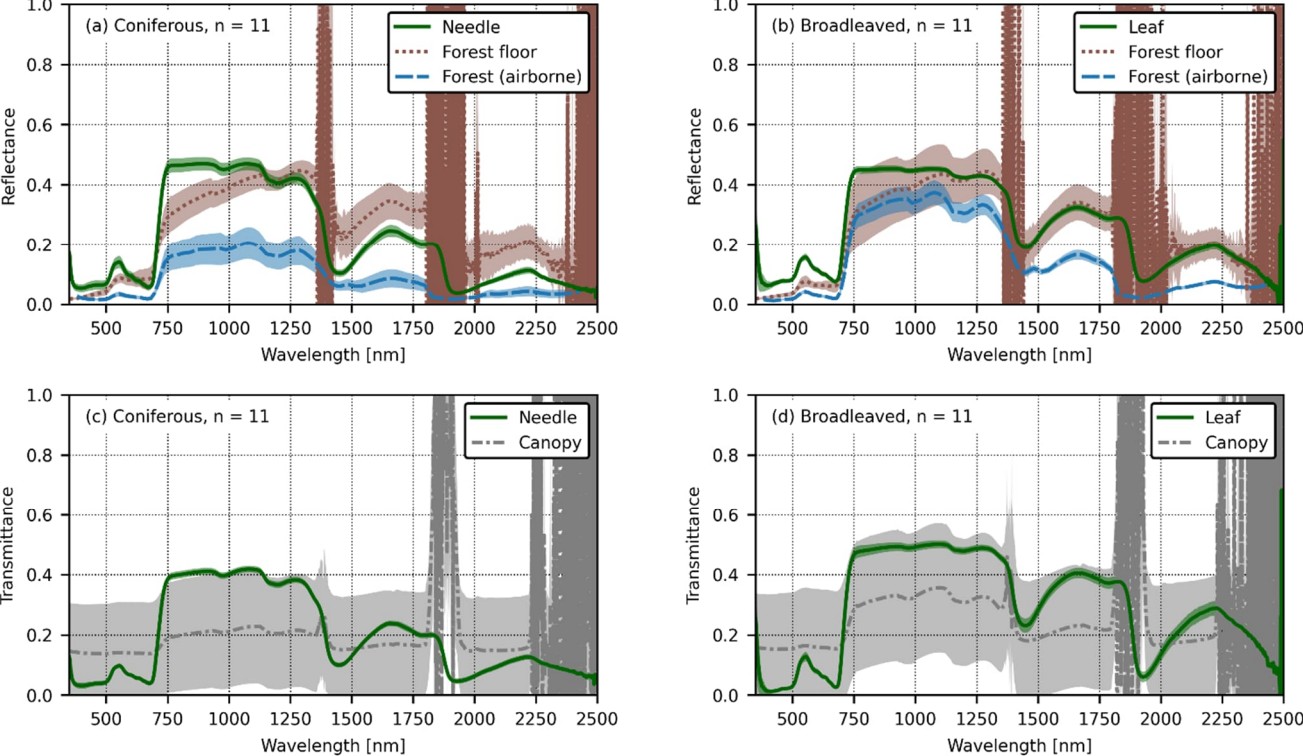


**Fig. 6.** Mean spectra at different scales. **A.** Mean reflectance spectra and their standard deviations for needles, forest floor and entire forest plot in coniferous forests. **B.** Mean reflectance spectra and their standard deviations for leaves, forest floor and entire forest plot in broadleaved forests. **C.** Mean transmittance spectra and their standard deviations for needles and canopies in coniferous forests. **D.** Mean transmittance spectra and their standard deviations for needles and canopies in broadleaved forests. The data shown in this figure are based on measurements and reflectance quantities described and defined in Sections 2.2.4-2.2.6 and 2.4.1., and only the subset of plots which had measurements of canopy transmittance are included here. Coniferous and broadleaved forests were defined so that at least 75% of the trees (based on basal area) within the plot were coniferous or broadleaved species, respectively. For visualization purposes, leaf-level reflectance and transmittance spectra were first computed at plot-level as averages weighted by tree species proportions and needle age classes, and then averaged over all plots to obtain the mean and standard deviation values shown in the above. Forest floor reflectance and canopy transmittance data are shown with the noise that is inherently present in atmospheric water absorption bands in spectral data measured outdoors. Forest reflectance (HDRF at plot-level, Fig. 6A and 6B) is averaged for an area of 25 m × 25 m in each stand, and is based on airborne CASI and SASI data.

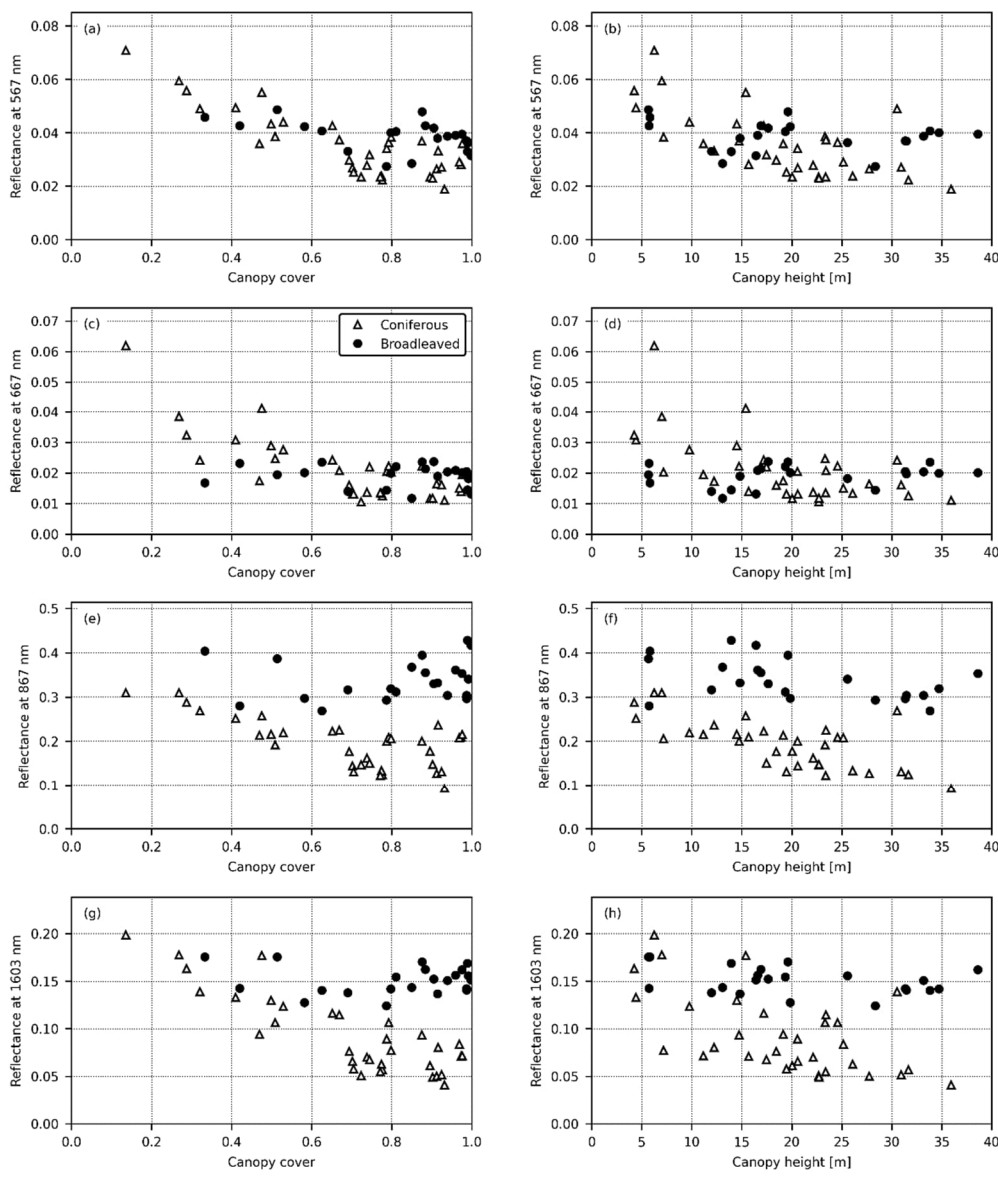

438

**Fig. 7.** The relationship between forest reflectance (HDRF, obtained from airborne CASI and SASI data) and forest structure (obtained from ALS data, scan zenith angle max 20°) for broadleaved and coniferous forests in four spectral regions: green (567 nm), red (667 nm), near-infrared (NIR, 867 nm) and shortwave infrared (SWIR, 1603 nm). The data are averaged for an area of 25 m × 25 m in each stand. Canopy cover was defined as the first echo cover index in ALS data, so that first echoes originating from the canopy were divided by all first echoes in the plot. **A. C. E. G.** Canopy cover and forest reflectance (HDRF). Spectral region indicated on the y-axis. **B. D. F. H.** Canopy height (defined as the 95th percentile of all canopy echoes in ALS data) and forest reflectance (HDRF). Spectral region indicated on the y-axis. Coniferous and broadleaved forests were defined so that at least 75% of the trees (based on basal area) within the plot were coniferous or broadleaved species, respectively. The data shown in this figure are based on measurements described in Section 2.4.



**4 Data availability**

The data are available in the open access repository Fairdata IDA which is a research data storage service provided by the Ministry of Education and Culture of Finland. The data can be accessed at: Hovi et al. 2024a https://doi.org/10.23729/9a8d90cd-73e2-438d-9230-94e10e61adc9 (for data described in Section 2.3.) and Hovi et al. 2024b https://doi.org/10.23729/c6da63dd-f527-4ec9-8401-57c14f77d19f (for data described in Section 2.4.).

**5 Conclusions**

Radiative transfer models of vegetation play a key role in advancing remote sensing science. The development of these models has been hindered by a lack of comprehensive ground reference data on both the structural and spectral characteristics of forests. In this paper, we introduced datasets containing information on the structural and spectral properties of temperate, hemiboreal, and boreal European forest stands. We anticipate that these data will have wide use in testing and validating radiative transfer models for forests and in other remote sensing studies beyond radiative transfer model development.

**Author contributions**

MR, AH and DS conceptualized the scientific data collection plan for the project. AH, DS, PL, ZL, LH and MR organized the field campaigns and participated in data collection or processing. JH was responsible for organizing the airborne operations and related data processing. AH curated the datasets and prepared data visualizations. MR prepared the manuscript with contributions from all co-authors. MR was responsible for project administration and funding.

**Competing interests**

The contact author has declared that none of the authors has any competing interests.

**Acknowledgements**

We thank Juho Antikainen, Lucie Červená, Petri Forsström, Bijay Karki, Jussi Juola, Titta Majasalmi, Eva Neuwirthová, Ville Ranta and Jaan Rönkkö for field work or data processing; Jan Pisek, Mait Lang, Mihkel Kaha and Andres Kuusk for support in organizing the measurement campaign in Estonia; Jana Albrechtová for resources in organizing the field measurements in the Czech Republic; Karel Holouš, Lukáš Fajmon and Tomáš Fabiánek for participation and support in airborne operations; Lucie Hradecká and Ilari Lähteenmäki for advice in data management planning; and staff of all field stations of our study sites for their help at different stages of the work.

**Financial support**

This study received funding from the European Research Council (ERC) under the European Union's Horizon 2020 research and innovation programme (grant agreement No 771049 / Rautiainen). The text reflects only the authors' view and the Agency



is not responsible for any use that may be made of the information it contains. The work of the Czech scientists was made
possible by the Ministry of Education of the Czech Republic, project LTAUSA18154: Assessment of ecosystem function
based on Earth observation of vegetation quantitative parameters retrieved from data with high spatial, spectral and temporal
resolution, and the CzeCOS program, grant number LM2023048.

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
