# Peer review of "A spectral-structural characterization of European temperate, hemiboreal and boreal forests"

_Earth System Science Data, 2024_

## Author Response (AR1)

Dear Editor,

We have provided our responses to the reviewers' comments in the following. Our responses are marked in green in the response letter and highlighted in yellow in the revised version of the manuscript. The line numbers in the response letter refer to the track-changes-version of the manuscript.

On behalf of all coauthors,
Kind regards,
Miina Rautiainen
* * *
**RC1**: **Anonymous Referee #1, 23 Jun 2024**

The development of radiative transfer (RT) models requires comprehensive ground reference data on both the structural and spectral characteristics of forests. However, such kind of data are currently lacking. The dataset introduced in this paper consists detailed structural data from forest inventory measurements, terrestrial and airborne laser scanning, and digital hemispherical photography, and optical properties of tree leaves and needles, forest floor and entire stands. Such data would be valuable for RT modeling and biophysical validation studies. The paper is finely prepared and I recommend its publication in ESSD.

- Thank you for the positive feedback on our manuscript!

It would be nice if some field measurement pictures can be provided.

- We added a set of photographs of the measurements in Appendix A. Please also note that the dataset itself (https://doi.org/10.23729/9a8d90cd-73e2-438d-9230-94e10e61adc9) includes several photos from each of the 58 study plots, so that the data users can get a quick understanding of what the forests looked like.

Fig. 7. The legend may be put in panel (a).

- We changed the location of the legend in Figure 7. It is now in panel (a).

**RC2: Anonymous Referee #2, 08 Jul 2024**

The study presents a new dataset based on forest inventory measurements with structural and spectral properties of 58 stands in temperate, hemiboreal and boreal European forests. The dataset is specifically designed to develop and validate radiative transfer models for forests but can also be utilized in other remote sensing studies. It includes detailed information on forest structure and spectral properties at multiple scales. The paper is well-written and structured and provides a valuable and comprehensive dataset for advancing RT models and remote sensing of forests, assuring FAIR data principles. However, I would like to ask the author for a few minor corrections:

- Thank you for the positive feedback on our manuscript!

L. 22: Specify these new satellite missions.

- We added examples of the new satellite missions (PRISMA, EnMAP, CHIME, SBG) (LINE 30).

Chapter 3: Does the data set contain information of measurement uncertainty? Please describe it in the manuscript, or give indications of how it can be derived.

- Thank you for this comment. Our dataset comprises very different types of data based on different measurement methods such as traditional forest inventory data collected with calipers and hypsometers, hemispherical photographs of tree canopies, laser scanning point clouds obtained with terrestrial and airborne sensors as well as spectrometer data measured in laboratory, field and from an aircraft. The data also represent different levels of processing. Thus, the uncertainties depend on the type of data and processing level and may comprise both sampling (biological) and measurement uncertainties. In general, in this dataset, the number of observations (per measurement method) are high and since all observations have been provided to the users in the dataset, standard errors or deviations can be calculated by the data users for their own application. In the Results section (Section 3), we also show the standard deviation of the relevant datasets in Figures 3, 4, 5 and 6 to give the users an idea of the variation. We have summarized the uncertainty estimates in the following and added this information also to the manuscript:
- In forest inventory measurements, typical uncertainties (standard deviation) in routine forest inventory measurements (diameter at breast height and tree height) are 0.3 cm and 0.5 m, respectively (https://doi.org/10.3390/f8020038). (LINE 145)
- In hemispherical photos, the systematic uncertainties in the gap fractions estimated from hemispherical photographs are less than 0.02. This estimate is based on a comparison of plot-level canopy gap fractions against canopy spectral transmittance measured at 450 nm for those plots in which canopy spectral transmittance measurements had been conducted. (LINE 187)
- In the terrestrial laser scanning, the uncertainties of the coordinates of individual points are in the order of millimeters. This estimate is based on the following: The calibration report provides the standard deviation of distance measurements as better than 1.2 mm, and that of angle measurement better than 8 arcseconds i.e. 0.002 degrees. The uncertainty of the angle measurement introduces an uncertainty to the coordinates of each point as 0.4 mm at a 10 m distance from the scanner, and 1.2 mm

at a 30 m distance. From these numbers it can be deduced that the uncertainties of the coordinates of individual points are in the order of millimeters. Furthermore, the terrestrial laser scanning dataset includes the registration reports from the Cyclone software which can be used to verify that the co-registration of the scans has the reported accuracy (i.e., better than 1 cm, as already mentioned in LINE 163).

- In spectral measurements of foliage, the maxima of uncertainty in general correspond to maxima of biological variation (https://doi.org/10.1016/j.rse.2021.112601). For coniferous (non-flat) needles, the errors are approximately 4%-6% in reflectance (DHRF) and 10%-12% in transmittance (DHTF) (https://doi.org/10.1109/JSTARS.2013.2292817). We added this information to the manuscript (LINE 305).
- In the spectral measurements of forest floor, we estimate that the individual measurements have an uncertainty of ~10% due to variation in illumination during the measurements (outdoors). Similarly, in the canopy spectral transmittance measurements, the uncertainty of canopy transmittance measurements (outdoors in clear-sky conditions) is ~5%. We added this information to LINE 217 and LINE 255.
- In the airborne hyperspectral data, the deviation between the atmospherically corrected airborne data and the ground-measured spectra of the reference targets was less than 2% across all wavelengths. This uncertainty of the atmospherically corrected CASI+SASI reflectance factors was estimated from reference targets that had their reflectance spectra measured on ground. Now added to LINE 339 in the manuscript.
- In the airborne laser scanning data, the lowest absolute positional accuracy was associated with coordinates of off-nadir points acquired at the edges of the flight lines (RMS 27 cm). The calculation of this accuracy is based on the performance of each component provided by manufacturer and an acquisition height of 1030 m above ground. Now added to LINE 354 in the manuscript.

L. 448 /Chapter 4: Data availability: Sharing scientific data effectively within the community is crucial, and I particularly appreciate your adherence to FAIR data principles, which greatly facilitates this process. However, FAIR principles are only mentioned and one reference is cited. Could you please give more details on how your data addresses FAIR principles, see also Box2 in Wilkinson et al., 2016 (https://www.nature.com/articles/sdata201618#Sec6)

- Thank you for this comment! Our dataset follows the guidelines (Findable, Accessible, Interoperable, Reusable) provided in Box2 in Wilkinson et al. 2016. The measurement data and detailed metadata files have been published in the Fairdata IDA Service and include information on the data usage licenses. Here is a summary of the Fairdata IDA Service:
*Fairdata IDA (ida.fairdata.fi) is a continuous service for safe research data storage, offered free of charge to Finnish universities and research institutes. The research dataset published have persistent identifiers (DOI). Data stored in IDA can be accessed with a browser or with command line tools. This makes the dataset findable for others and enables re-use of the data and creating a scientific reference. Data is frozen in IDA and stored in IDA in read-only mode, to minimize user errors. The integrity of the frozen data is further protected against loss or corruption by replicating each frozen file to a secondary storage medium and calculating an SHA256 checksum for each frozen file. Storing data in IDA allows published datasets containing multiple files and folder structures. A single dataset can include hundreds or even thousands of files and have a total volume of up to e.g. several terabytes, enabling users to make large datasets openly accessible with a single persistent*

*identifier.* [The text above was provided by CSC (source: https://www.fairdata.fi/en/ida/, accessed on 30 August 2024)].

- The "Data availability" section (Section 4, LINE 471) is very brief and follows the journal's instructions on how the section should be formulated. We added the requested information on FAIR principles in another section in the manuscript (LINE 75). It now reads: "The dataset follows the FAIR (Findable, Accessible, Interoperable, Reusable) principles."

**RC3**: **Anonymous Referee #3, 25 Jul 2024**

This manuscript presents a ground reference dataset of structural and spectral properties of 58 stands in temperate, hemiboreal and boreal European forests with objective to benchmark 3D radiative transfer models. As a radiative transfer model developer, I realize the importance of field measurements that helps the improvement of radiative transfer models and the space mission Cal/Val activities. I appreciate the presented work. Since I am not expert in field measurement, my comments and questions are mainly from a point of view of a modeler.

- Thank you for the positive feedback on our manuscript!

**Comments and questions:**

(1) Section 2.3: It is hard for readers outside of the domain to understand all the details of measurement. I suggest to add several schemata to present the ideas about the measured quantities and the geometry

- Thank you for mentioning this. We have now added a reference to a basic nomenclature paper which defines the measured reflectance quantities and their geometry in the context of optical remote sensing (Schaepman-Strub et al. 2006, added to LINE 107). The definitions provided in this paper are widely used in the spectral measurements community. In addition, we added the suggested schemata (Appendix A, Fig. A2) to show the illumination and view geometries during the measurements.

(2) Section 2.3.6: The reflection and transmission behaviour of leaves and trunks are usually anisotropic. Is it possible to measure BRDF (bi-directional reflectance distribution factor) and BTDF (bi-directional transmittance distribution factor) instead of DHRF and DHTF ? Also, some leaves have strong specular reflection, is the specular reflection included in DHRF ?

- The instruments that were available in this project (i.e., ASD RTS-3ZC integrating spheres) are designed to measure DHRF and DHTF and cannot produce data of the BRDF or BTDF of leaf or needle surfaces (or woody components). Measurements of leaf-level DHRF and DHTF are already rather laborious and slow, and multiangular measurements (BRDF or BTDF) would be even slower to make. In this study, we had a fairly large sample size (1314 leaf samples) and the advantage of this is that we are able to look at inter- and intraspecific variation in leaf-level spectral properties. If the measurements were slower (due to obtaining a multiangular data with another type of instrument), then also the sample size would have been significantly smaller. The answer to the second question is: yes, specular reflection was included in the DHRF. We have now added this detail to the manuscript (LINE 299).

(3) Section 2.3.6: Our radiative transfer model is able to simulate the uncertainty propagation (given the uncertainty of leaf optical properties, our model can compute the uncertainty of the simulated image/BRF), is it possible to get simultaneously the measurement and its uncertainty ? I downloaded the spectral transmittance through the link given by the authors, but I did not find information about uncertainty.

- We are happy to hear that you have already tested our data for leaf-level transmittance! Please see our response to reviewer #2 regarding the uncertainties of the data.

(4) Section 2.4.1: For the atmospheric correction, how do you get the local atmospheric profile ?

- Parameters used in the atmospheric corrections were retrieved directly from the airborne hyperspectral data. However, for the Hyytiälä site, parameters were specifically retrieved from the on-site Aeronet station (CIMEL sunphotometer). We have now specified this in the manuscript (LINE 336).

**RC4**: **Anonymous Referee #4, 08 Aug 2024**

The validation of BRDF models and inversion methods are in urgent need of comprehensive ground reference data of forests, as data measurement is extremely difficult. This manuscript proposes a whole dataset on the structural and spectral properties of 58 stands in temperate, hemi-boreal and boreal European forests. The dataset contains both forest structure data and spectral properties at leaves and needles, tree, forest floor and entire stands scales base on in situ and airborne measurements. This dataset is valuable. But the manuscript can be further improved. I suggest the acceptance after revisions with clarification of minor issues.

- Thank you for the positive feedback on our manuscript!

1) Line 120, It's better to describe the exact location of "six fixed locations".

- The" six fixed locations" where the photos were taken were all four corners of the plot and in the center of the plot (in two different directions). We added a sentence explaining this (LINE 106).

2) Line 127, It's look like there's a writing mistake as "it would be 8cm if h > 16m", or maybe can explain why the thresholds were determined as this. This is in conflict with the previous definition of mature stands (Line 122).

- Thank you for spotting the typo! It indeed should say if $h > 16$ m (and not "if $h < 16$ m", as we had originally written). This has now been corrected (LINE 131).

3) Line 264, What are the criteria for determining whether a needle is current year or one-year-old needle, according to color?

- Needle age classes were not determined only based on color. The position on a branch, and the color of needles and bark of the shoot are macroscopic criteria that were used to recognize current year shoots and needles. They are usually found in the terminal branch positions in sun-exposed branches. The previous-year shoot can be recognized by the presence of dead and partly shed bud scales at the base of the current year shoot. This rule can be used retrospectively to assess shoot and needle age in regularly growing shoots. More information on the developmental process of forming shoots from various bud categories can be found in e.g., Polák et al. 2007 (DOI 10.1007/s00468-006-0093-z). We added a comment on this to the manuscript (LINE 275).

4) Formula (3), What is the measurement method of $S_{ref,T}$ ? According to the formula, $S_{ref,T}$ in the formula seems to have the same numerical value as $S_{ref,R}$? Please clarify it.

- The white reference measurements for reflectance ($S_{ref,R}$) and transmittance ($S_{ref,T}$) are not the same, because the reflectance and transmittance measurements in the ASD RTS-3ZC integrating sphere are made in two different configurations, utilizing different sample ports of the integrating sphere. The use of two separate configurations is according to the manufacturer's instructions. An illustration of the reflectance and transmittance measurement configurations has been given in Fig. S1.1 in the publication of Hovi et al. (2020) (https://doi.org/10.14214/sf.10270).

5) Line 419 (Fig. 5.) What is the meaning of n in the picture (Number of trees?) (n=34. n=22)? Same problem also can be found in Line 124 and Fig.6.

- "n" refers to the number of plots. This has now been specified in all the mentioned locations (Fig. 5 & 6 captions, and LINE 128).

6) Fig.7. Why only (c) has a legend?

- Per request from Reviewer #1, we changed the location of the legend in Figure 7. It is now in panel (a). The same legend is valid for all subfigures.